# Differences in the perception of harm assessment among nurses in the patient safety classification system

**Kwangmi Lee**[1]☯**, Kyeongsuk Yoon**[2]☯**, Byeongsook Yoon**[2]☯**, Eunhee Shin**[3]*

**1** Department of Nursing, National Cancer Center, Goyang, Republic of Korea, **2** Division of Nursing, Yonsei University Wonju Severance Christian Hospital, Wonju, Republic of Korea, **3** Department of Nursing Science, Sangji University College of Health Sciences, Wonju, Republic of Korea

☯ These authors contributed equally to this work.
* hshin1970@sangji.ac.kr

## Abstract

### Background

Precise harm assessment by the medical staff is very important in a patient safety event reporting system but there are differences in perception due to insufficiencies in education.

### Methods

We developed the survey tool consisting of nine patient safety incident scenarios to investigate the interrater agreement in the harm score assigning among nurses. The survey tool was distributed to 287 nurses working at two hospitals.

### Results

The overall kappa value for interrater agreement was $k = 0.21$ for harm and $k = 0.28$ for harm duration. In nine patient safety event scenarios, such as "mislabeled specimen" or "chest tube drain", when the degree of harm was not clear, the assessments of harm and harm duration were somewhat dispersed.

### Conclusion

For the quality of the patient safety incident reporting system, the accurate harm assessment of medical personnel is highly important; however, results in this study indicated that theassessment of the degree of harm by Korean nurses was not standardized. The reason for this variability could be due to the lack of education that takes harm assessment into account. Therefore, training in harm assessment and the development of programs to support this training are both necessary.

**Data Availability Statement:** All relevant data are within the manuscript and its Supporting information files.

**Funding:** This study was supported by the NRF (National Research Foundation of Korea) funded by

the MOE(Ministry of Education) (NRF-
2018R1D1A3B07049955). The funders had no role
in study design, data collection and analysis,
decision to publish, or preparation of the
manuscript.

**Competing interests:** No authors have competing
interests.

## Introduction

In 2003, the U.S. National Institute of Medicine for 'Patient Safety: Achieving a New Standard of Care' stated that in order to reduce the number of preventable medical accidents, information regarding patient safety, including near misses and adverse events, needs to be standardized and managed [1]. Nevertheless, due to differences in patient safety event reporting systems, differences in the definitions of accidents, measurements of their frequencies, and the tracking of patient safety events have been reported [2]. Heterogeneous data entry systems, differences in terminology and classification of patient safety, and inconsistent characteristic usage in patient safety event reporting systems make standardization difficult; furthermore, it has been reported that the analysis and interpretation of near misses and adverse events in each data set require different individual methods [3].

In order to make clear and direct efforts for patient safety, healthcare professionals rely on accurately described information regarding patient safety events, harm of event and the main causes of the event. Among these, the allocation of harm scores is a crucial step in the study of patient safety events, as it often provides an opportunity to trigger a response to the management of harm and the event on an organizational level [4]. Regarding patient safety events, the management of adverse events or sentinel events causing direct harm to patients in particular has recently become the main capability of relevant professionals [4].

Studies regarding harm of patient safety events were hardly performed until the beginning of the 1960s, and only deviations from clinical guidelines were examined in relation to preventable harm. In examining such harm, it is known to be very difficult to determine whether the harm originated from medical treatment, whether the medical treatment was unnecessary, or the harm was preventable. Consequently, from a patient-to-health care provider's perspective, a variety of efforts such as the classification of harm from the occurrence of an event and the exact assessment are being continuously made [5].

To date, harm classification schemes have been developed by organizations such as the World Health Organization (WHO), the National Quality Forum (NQF), the National Coordinating Council for Medication Error Reporting and Prevention (NCC MERP), and the Agency for Healthcare Research and Quality (AHRQ) [4–6]. In particular, the National Coordinating Council for Medication Error Reporting and Prevention (NCC MERP) created its own index classifying medication errors by elaborating the definitions of harm levels in 2001. Moreover, in 2010, Version 1.1 of the Common Format Harm Scale was presented by the Agency for Healthcare Research and Quality (AHRQ) according to the Patient Safety and Quality Improvement Act of 2005, which requires all healthcare providers to employ a common set of data elements when reporting patient safety events for the purpose of data statistics and analysis, and to use the Patient Safety Organization's (PSO's) standardized method to collect and classify data. This version categorized the impact of patient safety events on patients' functional capacities on a seven-point scale, categorized into five categories of "No Harm" and "Death," and Version 1.2 was released in April 2012, allowing users to classify harm on a five-point scale from no harm to death [4].

Despite the widespread use of these various patient safety event reporting systems in medical institutions, factors that hinder consistent, timely and accurate reporting have been reported. First, factors that affect medical personnel who use the reporting system include the level of clarity about the purpose of the report, the usefulness of the system, and the degree to which reporting is emphasized in institutional communication [7, 8]. In addition, it is influenced by the reporter's understanding and interpretation of the definition of harms and contributing factors, and the reporter's perception and personal judgment, along with the organization's safety culture [7, 9, 10]. In particular, for an accurate patient safety event

reporting system, consistency is important in the definition and classification of near misses and adverse events of nurses and other medical personnel who enter data into the system [9, 11]. In many studies, it was reported that there were inconsistent results in the harm assessment in the report of patient safety events between medical staff [7, 10, 12–14]. For the consistency of risk assessment, the need for related education programs was raised [10, 11].

In Korea, the Patient Safety Act was implemented and entered in force in 2015, and in accordance with the Patient Safety Act, healthcare workers and patients who have caused or become aware of patient safety events can report them autonomously, and the medical institution implements and operates a system for reporting and learning of patient safety accidents for investigation, research and sharing of such self-reporting contents. In the current patient safety event report learning system, there are general reports and intensive reports. In the general report form, the date and time of the event, the date of discovery, and the improvement measures to prevent the recurrence of the event are entered, and in the intensive report form, the degree of harm should be assessed along with the date and time of the event. Among these, the voluntary reporting of patient safety events and a standardized assessment of the harm degree of the event and contributing factors are strongly needed to manage patient safety events and promote qualitative improvement based on reported events.

While the accurate assessment of harm by medical staff is highly important, factors such as lack of sufficient training contribute to differences in perception; therefore, in this study, a harm assessment tool comprising nine patient safety scenarios for nurses working in medical institutions was applied to evaluate the degree of differences in the perception of harm among nurses.

## Materials and methods

### Research design

The present study is an exploratory study to determine the degree of agreement in harm assessment among nurses working in national cancer centers and advanced general hospitals in different regions using an assessment tool for the harm of patient safety events.

### Subject of study and data collection from subjects

The number of subjects required varies from the minimum reported by Mundfrom et al. [15], according to whom an absolute number of at least 100 study subjects are needed for factor analysis, to studies reporting that at least 3 to 20 times the number of factors are needed. However, there is to date no consensus on the size of the research sample adequate for exploratory factor analysis. Therefore, as 287 nurses participated in the actual survey of this study, a total of 287 subjects were included in the analysis.

Data were collected using self-report questionnaires conducted to nurses working in National Cancer Center from March 5 to March 14, 2018 and Yonsei University Wonju Severance Christian Hospital from June 25 to 5 July, 2019. The subjects who wished to voluntarily participate in this study via recruitment notice received a written explanation about this study and completed the questionnaire after agreeing to the consent form.

### Research tools

**Instrument.** The questionnaire was developed with reference to the content of the study conducted by Tamara et al. [4] with the Common Format Harm Scale Version 1.2 of the AHRQ and was configured to assess harm and harm duration for each of nine patient safety incidents scenarios: (1) medication given via wrong route, (2) body part laceration during

surgery, (3) contrast allergy, (4) abdominal infection, (5) mislabeled specimen, (6) wrong site surgery, (7) chest tube drain, (8) medication overdose, and (9) medication given at the wrong time. However, we organized the patient safety incident scenarios from the "Patient Safety Event Cases and Prevention" [16], published by the Korean Hospital Nurse Association to better reflect the real-world incidents in Korea. After completing the scenarios, three experts working in patient safety departments at medical institutions and one university professor reviewed and examined whether the selected scenarios were appropriate, after which this study was performed.

The questionnaire also included 13 descriptive questions to determine each subject's gender, current department, working period at current job, job satisfaction at current job, Whether or not getting a patient safety training at current job, method of a patient safety training, contents of a patient safety training, time patient safety training, whether or not participating events of a patient safety at current job, experience of patient safety incident at current job, type of patient safety incident, reporting on patient safety incident, types of patient safety incidents experienced.

**Definition of harm and harm duration of questionnaire.** In this study, the assessment of harm and harm duration for domestic patient safety events in each scenario was based on the following classification by applying the criteria [17] suggested by NCC MERP. The harm was categorized as near miss (A): an circumstances or events that have the capacity to cause error, such as disorganized medical equipment; near miss (B): an error occurred but the error did not reach the patient; no harm (C): an error occurred that reach the patient but did not cause patient harm; mild harm (D): an error occurred that reached the patient and required monitoring to confirm that it resulted in no harm to the patient and/or required intervention to preclude harm; Moderate harm (E): an error occurred that may have contributed to or resulted in temporary harm to the patient and required intervention; Severe harm (F): an error occurred that may have contributed to or resulted in permanent patient harm and required the short-term or long-term hospitalization; and a sentinel event: an error occurred that may have contributed to or resulted in the patient's death. Harm duration was categorized as permanent: an error occurred to the patient persisting over a year; temporary: an error occurred to the patient persisting less than a year; and unknown.

## Ethical considerations

To ensure ethical consideration of the subjects, this study was approved by the Ethical Committee of National Cancer Center (IRB No.: NCC 2018–0033) and Sangji University (IRB No.: IRB 46). The researcher provided a written explanation including the purpose of the study, voluntary participation, and their anonymity, and a consent form for nurses who wished to voluntarily participate in the study. When the subject completed the consent form, the researcher requested that the subject complete the questionnaire.

## Data analysis

Collected data were analyzed using SPSS 26.0 (SPSS Inc, Chicago, IL, USA). The general characteristics of the study subjects were presented as frequency and percentage, and the frequency of respondents was calculated for each scenario. In addition, Fleiss' kappa was calculated by estimating the interrater agreement for each scenario among respondents. A standard statistical analysis of the kappa values was used. The levels of agreement were almost perfect when kappa had a value of 0.81–1.00; a substantial agreement when it was 0.61–0.8; a moderate agreement when it was 0.41–0.60; a fair agreement when it was 0.21–0.40; a slight agreement when it was 0.01–0.20; and 0 or less was interpreted as virtually no agreement.

**Table 1. General characteristics of study subjects.**

| Variables | Categories | n (%) |
|---|---|---|
| Gender | Male | 15(5.2) |
| | Female | 271 (94.4) |
| | Missing | 1(0.3) |
| Current department | General ward | 192 (66.9) |
| | Special ward (ICU, ER, etc) | 77(26.8) |
| | Outpatient department | 10(3.5) |
| | Others+ | 7(2.4) |
| | Missing | 1(0.3) |
| Working period at current job (year) | <1 | 9(3.1) |
| | ≥1-<5 | 136 (47.4) |
| | ≥5-<10 | 64(22.3) |
| | ≥10 | 76(26.5) |
| | Missing | 2(0.7) |
| Job satisfaction at current job | Very satisfied | 10(3.5) |
| | Satisfied | 184 (64.1) |
| | Unsatisfied | 80(27.9) |
| | Very unsatisfied | 12(4.2) |
| | Missing | 1(0.3) |
| Whether or not getting a patient safety training at current job | Yes | 283 (98.6) |
| | No | 3(1.0) |
| | Missing | 1(0.3) |
| Method of a patient safety training (duplication check) | Theoretical lectures | 217 (75.6) |
| | Case-based discussion training | 40(13.9) |
| | Certificated brochure training | 196 (68.3) |
| | Department conveying training | 191 (66.6) |
| | Others++ | 15(5.2) |
| Contents of a patient safety training (duplication check) | Understanding Patient Safety | 261 (90.9) |
| | Time and method of patient identification | 247 (86.1) |
| | Grade and criteria of patient safety incident reporting | 180 (62.7) |
| | Incident reporting procedures | 257 (89.5) |
| | Inpatient care management | 148 (51.6) |
| | Activating near miss reporting | 194 (67.6) |
| | Others | 1(0.3) |
| Time of a patient safety training (hour) | <1 | 44(15.3) |
| | ≥1-<4 | 170 (59.2) |
| | ≥4-<8 | 36(12.5) |
| | ≥8 | 35(12.2) |
| | Missing | 2(0.7) |

*(Continued)*

**Table 1.** (Continued)

| Variables | Categories | n (%) |
|---|---|---|
| Whether or not participating events§ of a patient safety at current job | Yes | 150 (52.3) |
| | No | 133 (46.3) |
| | Missing | 4(1.4) |

[+]Operating room,

[++] Cyber training, practical training,

§ Special lecture, seminar, Campaign, etc.

## Results

### General characteristics of subjects

A total of 287 subjects were included in this study and 272 (94.8%) were female. The subjects' working departments included 192 (66.92%) general wards, 77 (26.8%) special departments such as intensive units and emergency departments, 10 (3.5%) outpatient departments. The largest group, consisting of 136 subjects (47.4%), had worked at least 1 year and less than 5 years in their current job; 76 subjects (26.5%) had worked more than 10 years; 64 subjects (22.3%) had worked at least 5 years and less than 10 years; and 9 subjects (3.1%) had worked less than a year. Of the subjects, 194 (67.6%) reported they were satisfied with their current jobs. Currently, 283 subjects (98.6%) reported that they had received the patient safety training at work. In regard to the training methods for patient safety with duplicate answers, 217 subjects (75.6%) had trained with the "theoretical lectures", followed by 196 (68.3%) with "certificated brochure training" and 191 (66.6%) with "department conveying training". As for the contents of patient safety training, 261 (90.9%) reported the "understanding of patient safety", and 257 (89.5%) reported the "incident reporting procedure". The number of subjects who spent more than one hour and less than four hours on patient safety training was the highest at 170 (59.2%). Regarding the participating events of a patient safety such as special lectures, seminars, and campaigns at their current workplace, 150 subjects (52.3%) reported that they had participated (Table 1).

### Subjects' patient safety incident experiences

When asked about their experiences with patient safety incident at current job, 207 subjects (72.1%) reported they had experienced patient safety incidents at work. Regarding the types of patient safety incidents that they had experienced, 147 subjects (51.2%) reported with near miss and 131 subjects (45.6%) reported with no harm events as duplicate answers, while 61 subjects (21.3%) reported with adverse events and 22 subjects (7.70%) reported with sentinel events. When asked whether they had written a report regarding the patient safety incidents, 171 (59.6%) of the subjects reported that they had. In the question about the type of patient safety incidents that they had experienced, which allowed multiple answers, 152 subjects (53.0%) reported with medication errors, followed by falling, answered by 124 subjects (43.2%) (Table 2).

### Consistency rate among subjects regarding the scenarios

In response to each scenario, 135 subjects responded "mild harm (D)" to the scenario "medication given via wrong route" and 74 subjects responded "moderate harm (E)". Regarding the

**Table 2. Experience of patient safety accident in study subjects.**

| Variables | Categories | n(%) |
|---|---|---|
| Experience of patient safety incident at current job | Yes | 207 (72.1) |
| | No | 77(26.8) |
| | Missing | 3(1.0) |
| Type of patient safety incident (duplication check) | Near miss | 147 (51.2) |
| | No harm safety event | 131 (45.6) |
| | Mild/Moderate/Severe safety event | 61(21.3) |
| | Sentinel event | 22(7.7) |
| Reporting on patient safety incidents | Yes | 171 (59.6) |
| | No | 38(13.2) |
| | Missing | 78(27.6) |
| Types of patient safety incidents experienced (duplication check) | Surgery | 11(3.8) |
| | Delivery | 0(0.0) |
| | Treatment procedure | 20(7.0) |
| | Anesthesia | 1(0.3) |
| | Clinical examination | 22(7.7) |
| | Blood transfusion | 3(1.0) |
| | Medication | 152 (53.0) |
| | Infection | 4(1.4) |
| | Computerized disorder | 7(2.4) |
| | Medical equipment/Medical device | 15(5.2) |
| | Hospital meal | 20(7.0) |
| | Fall | 124 (43.2) |
| | Treatment material contamination /failure | 2(0.7) |
| | Suicide/Self-harm | 18(6.3) |
| | Other | 2(0.7) |

harm duration of the same scenario, 128 subjects responded "temporary", but 146 responded "unknown." In the second scenario, "body part laceration during surgery", 151 subjects evaluated it as a "severe harm (F)" and harm duration was also rated the highest, with 153 subjects responding "permanent". To the third scenario "contrast allergy", 131 subjects responded "moderate harm (E)" and 115 subjects responded "severe harm (F)", while regarding harm duration, 211 subjects answered "temporary". The fourth scenario, "abdominal site infections" was assessed by 143 subjects as "severe harm (F)" and by 95 subjects as "sentinel event". One hundred twenty-five subjects responded to the harm duration of this scenario with "permanent" but 97 subjects answered "unknown". The degree of harm in the fifth scenario, "mislabeled specimen" was assessed by 110 subjects as a "moderate harm (E)" and by 103 subjects as a "sentinel event"; regarding the harm duration, 144 subjects evaluated it as "temporary". The harm degree of the sixth scenario, "wrong site surgery" was evaluated by 203 subjects as a "sentinel event" and the harm duration was rated as "permanent" by 271 subjects. The harm degree of the seventh scenario, "chest tube drain" was rated by 121 subjects as a "mild harm (D)" and

**Table 3. Agreement values for harm scale and harm duration.**

| Scenario | Harm Score: Frequency of Respondents (%) | | | | | | | Harm Duration: Frequency of Respondents (%) | | | kappa value [a] | |
|---|---|---|---|---|---|---|---|---|---|---|---|---|
| | Near Miss (A) | Near Miss (B) | No Harm (C) | Mild Harm (D) | Moderate Harm (E) | Severe Harm (F) | Sentinel Event | Permanent | Temporary | Unknown | Harm score | Harm duration |
| 1. Medication given via wrong route | 19 (6.6) | 9 (3.1) | 22 (7.6) | 135 (46.9) | 74 (25.7) | 16 (5.6) | 11 (3.8) | 12 (4.2) | 128 (44.4) | 146 (50.7) | 0.483 | 0.575 |
| 2. Body part laceration during surgery | 0 (0.0) | 1 (0.3) | 0 (0.0) | 2 (0.7) | 79 (27.4) | 151 (52.4) | 54 (18.8) | 153 (53.3) | 75 (26.1) | 58 (20.2) | 0.526 | 0.537 |
| 3. Contrast allergy | 3 (1.0) | 2 (0.7) | 8 (2.8) | 17 (5.9) | 131 (45.6) | 115 (40.1) | 9 (3.1) | 18 (6.3) | 211 (73.5) | 56 (19.5) | 0.527 | 0.661 |
| 4. Abdominal site infection | 7 (2.4) | 9 (3.1) | 0 (0.0) | 13 (4.5) | 17 (5.9) | 143 (49.8) | 95 (33.1) | 125 (43.6) | 63 (22.0) | 97 (33.8) | 0.522 | 0.512 |
| 5. Mislabeled specimen | 1 (0.3) | 2 (0.7) | 2 (0.7) | 2 (0.7) | 110 (38.3) | 66 (23.0) | 103 (35.9) | 47 (16.4) | 144 (50.2) | 95 (33.1) | 0.496 | 0.532 |
| 6. Wrong site surgery | 0 (0.0) | 2 (0.7) | 0 (0.0) | 1 (0.3) | 8 (2.8) | 72 (25.1) | 203 (70.7) | 271 (94.4) | 7 (2.4) | 8 (2.8) | 0.645 | 0.914 |
| 7. Chest tube drainage | 3 (1.0) | 19 (6.6) | 51 (17.8) | 121 (42.2) | 82 (28.6) | 6 (2.1) | 4 (1.4) | 4 (1.4) | 244 (85.0) | 38 (13.2) | 0.481 | 0.778 |
| 8. Medication overdose | 5 (1.7) | 10 (3.5) | 10 (3.5) | 138 (48.1) | 92 (32.1) | 22 (7.7) | 8 (2.8) | 11 (3.8) | 176 (61.3) | 99 (34.5) | 0.509 | 0.601 |
| 9. Medication given at the wrong time | 10 (3.5) | 54 (18.8) | 91 (31.7) | 107 (37.3) | 18 (6.3) | 6 (2.1) | 1 (0.3) | 2 (0.7) | 153 (53.3) | 132 (46.0) | 0.469 | 0.596 |
| Overall kappa value [b] (Harm scale) | | | | | | | | | | | 0.205 | |
| Overall kappa value [b] (Harm duration) | | | | | | | | | | | 0.279 | |

[a] Fleiss' kappa calculated by adding option standard variables.

[b] Index of agreement with Fleiss' Kappa.

Adapted from Williams T, et al. The reliability of AHRQ Common Format Harm Scales in rating patient safety events. J Patient Saf 2015 Mar;11(1):52–9.

by 82 subjects as a "moderate harm (E)"; the harm duration was rated by 244 subjects as "temporary." In the eighth scenario, "medication overdose", 138 subjects responded "mild harm (D)"and 92 subjects responded "moderate harm (E)" and regarding the harm duration, 176 subjects answered "temporary." With regard to the harm degree of the last scenario, "medication given at the wrong time", 107 subjects responded "mild harm (D)" and the harm duration was rated "temporary" by 153 subjects.

Upon analysis of the interrater agreement rate among the respondents using Fleiss' kappa, the harm evaluation yielded a value of 0.21 and the value for harm duration was 0.28. (Table 3).

## Discussion

Our findings demonstrate that the overall interrater agreement for harm of patient safety incidents is 0.21 in this study, which is significantly lower than the result of the similar studies using AHRQ Common Format Ham Scale Version 1.2 [4, 18]. In the study conducted by Tamara et al. [4], the value of interrater agreement was 0.45, and other study conducted by Toni et al., the value was 0.48 [4, 18]. In the general characteristics of the study subjects, most of subjects of this study are nurses who have experience in their current workplace for more than one year and less than five years, and who have received patient safety training for more

than one hour and less than four hours. While the study subjects of the research conducted by Tamara et al. [4] were quality, risk and safety managers, and thus had more likely to have an experience in assessing the harm of patient safety incidents. However, the subjects in the study by Toni et al. [18] had 5 years of experience or less which is similar to the characteristics of the subjects in our study. As the patient safety reporting system was activated in Korea in accordance with the relevant laws in 2015, it is possible that nurses in our study had less experience in harm assessment than those in previous studies but further research is needed in the future considering other factors.

Also, the reason for the low interrater agreement in this study could have been subjective judgments about harm scores because the description of harm scores was only provided with patient safety scenarios in the questionnaire. Another reason for the low interrater agreement rate in the harm assessment are that the level of understanding of the reporter's harm scale is reported to affect the assessment [7]. When harm assessments were conducted for nine scenarios involving patient safety incidents in this study, a generally moderate agreement was shown when the harm was known, such as "body part laceration during surgery" or "wrong site surgery". On the other hand, in cases in which the assessment of harm was unclear, such as "mislabeled specimen" or "chest tube drain" the evaluation of harm and harm duration were shown to be somewhat dispersed.

As such, the need for education among reporter is raised as a more accurate harm assessment method. In many studies including Pronovost et al., professional training and sharing of ideas in practice have been reported to help enhance the perception of medical personnel in patient safety incident scenarios [7, 10, 14, 19].

In the results of harm assessment for patient safety incidents for nurses with similar experience, the interrater agreement value of this study was significantly lower than that of other studies. Therefore, it is necessary to develop a training program in Korea to actively assist nurses in performing their patient safety duties through continuous monitoring of patient safety reports and practical training on harm assessment in reporting patient safety incidents. It is also required that the development of multidisciplinary, academic tools for the standardization of harm assessment.

Despite the fact that such research was first conducted in Korea, where much attention has been paid to patient safety along with the establishment of a patient safety reporting system, as a main limitation of this study, there is a lack of generalizability since only nurses working at two medical institutions were surveyed. Next, in terms of validity of the questionnaire, we only performed the content validity with three experts without pilot study, face validity and criterion validity.

In addition, the study involved working nurses based on the assumption that they had already received the respective primary training, so no explanation or separate training regarding harm or harm duration assessment was provided. Consequently, since the nurses were prompted to read and evaluate each scenario in the questionnaire immediately, individual differences might have affected the interpretation of the scenario and assessment, and differences in individual education and clinical experience might be reflected in their assessment.

## Conclusion

In the patient safety incident reporting system, the accurate assessment of medical personnel is highly important; however, there are differences in perception due to factors such as inadequate training. Therefore, in the present study, AHRQ Common Format Ham Scale Version 1.2 was applied to identify differences in perception of harm assessment among nurses. This study analyzed the interrater agreement among respondents via Fleiss' kappa, for which a

value of 0.21 was found for harm assessment and 0.28 for harm duration. These values indicated significantly lower consistency in the harm assessment among nurses.

For the quality of the patient safety incident reporting, it is important for healthcare professionals to accurately assess the degree of patients' harm. This study found that the assessment of the degree of harm by Korean nurses was not standardized. The reason for this variability could be due to the lack of education that takes harm assessment into account. Therefore, training in harm assessment and the development of programs to support this training are both necessary.

## Supporting information

**S1 Questionnaire. (English version).**
(PDF)

**S2 Questionnaire. (Korean version).**
(PDF)

**S1 Dataset.**
(XLSX)

## Acknowledgments

The contributions of all participants in this study are greatly appreciated. We would like to thank Editage (www.editage.co.kr) for English language editing.

## Author Contributions

**Conceptualization:** Kwangmi Lee, Kyeongsuk Yoon, Byeongsook Yoon, Eunhee Shin.

**Data curation:** Kwangmi Lee, Kyeongsuk Yoon, Byeongsook Yoon, Eunhee Shin.

**Formal analysis:** Eunhee Shin.

**Funding acquisition:** Eunhee Shin.

**Investigation:** Kwangmi Lee, Kyeongsuk Yoon, Byeongsook Yoon, Eunhee Shin.

**Methodology:** Kwangmi Lee, Kyeongsuk Yoon, Byeongsook Yoon, Eunhee Shin.

**Project administration:** Eunhee Shin.

**Resources:** Eunhee Shin.

**Software:** Eunhee Shin.

**Supervision:** Eunhee Shin.

**Validation:** Kwangmi Lee, Kyeongsuk Yoon, Byeongsook Yoon, Eunhee Shin.

**Writing – original draft:** Eunhee Shin.

**Writing – review & editing:** Kwangmi Lee, Kyeongsuk Yoon, Byeongsook Yoon.

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
