## [Decision Letter · Decision Letter 0]

11 Aug 2020

PONE-D-20-19403

Differences in the perception of harm assessment by nurses in the patient safety classification system

PLOS ONE

Dear Dr. Shin,

Thank you for submitting your manuscript to PLOS ONE. After careful consideration, we feel that it has merit but does not fully meet PLOS ONE’s publication criteria as it currently stands. Therefore, we invite you to submit a revised version of the manuscript that addresses the points raised during the review process.

Specially, the reviewers have questioned the validity and reliability of your data collection tools and related process. You know that an unreliable and invalid questionnaire does not lead to valid and reliable data. Also, in such a condition, the accuracy of collected data and its interpretation are under question. Therefore, you are required to answer the reviewers’ comments one by one and incorporate changes with full details. Next, I will consider your article for another review round.   

We look forward to receiving your revised manuscript.

Kind regards,

Prof, Mojtaba Vaismoradi, PhD, MScN, BScN

Academic Editor

PLOS ONE

Journal Requirements:

2. In your financial disclosure, please clearly specify whether the funders played any role in the study.

3.Please provide further details regarding how participants were recruited, including the participant recruitment date.

4. Please clarify in the methods section where the nurses were recruited from (which hospital/study sites).

5. Thank you for providing the Korean version of the questionnaire as Supporting Information. Please also include an English copy as Supporting Information.

6. Thank you for including your ethics statement:

"To ensure ethical consideration of the subjects, approval by the Institutional Review Board (IRB) of the institutions to which the study subjects belonged was received after review before conducting study, and the study was performed after explaining the purpose of the study and asking for cooperation to the hospital's nursing headquarters. The researcher provided a written explanation including the purpose of the study, voluntary participation, and anonymity, and a consent form for nurses who wished to voluntarily participate in the study via a recruitment notice; in case of voluntary consent, the consent form was signed, and when the subject completed the consent form, the researcher requested that the subject complete the survey."

Reviewers' comments:

Reviewer #1: In this study, the authors attempt to assess the differences in nurses' perceptions of harm using scenarios. While this is an interesting study, I have concerns about the following:

According to the author, this study is based on the fact that accurate assessment of harm by healthcare providers is highly important, and that factors such as lack of adequate training can lead to differences in perceptions of harm. However, there is no citation from the literature on this in the Introduction section.

The relationship between the importance of standardization of classifications and terminology in incident reporting, which the author mentions in the Introduction and at the beginning of the Discussion, and the differences in nurses' perceptions of events (particularly in terms of impact on patients and duration of harm) that this study tried to identify is unclear. I think that this lack of clarity can be attributed to the lack of reference to how differences in nurses' perceptions of harm may be detrimental to incident reporting.

This problem also affects the Discussion. The first paragraph of the Discussion is what should be stated in the Introduction, and the second paragraph just redefines what has already been defined in the Methods. In the Discussion, starting from the third paragraph, the results of this study, the following sections have not been adequately discussed: 1) General Characteristics of subjects, 2) Subjects' patient safety event experiences, and 3) Consistency Rate among subjects regarding the scenarios. This may be because the purpose of this study is not properly established in the Introduction by citing the literature. Therefore, the authors did not compare the results of this study to the findings of previous studies and show what new findings this study represents in this research area. The paragraph from line 228 suddenly refers to education, but the authors should mention more about the relationship between education and this study in the Introduction.

In summary, it is necessary to clearly state what this study tries to identify in the field of incident reporting and patient safety and to state that this requires a method of determining the rate of agreement between the nurses' perceptions of harm based on the scenarios. The results obtained then need to be discussed in relation to previous studies.

Minor comments

p.7, line 139 - I think the author forgot to delete the phrase, "Contents of a patient safety training."

p.7, line 141 - Please make sure the phrase "the number of patients." is correct.

p.9, line 157 - Please define "patient safety event."

p.9, line 163 - It is difficult to understand the phrase, "medication error was the most common answer ".

p.10, line 189 - Please use the word for Fleiss' assessment of the Kappa value in the Method section of line 123. "Showing a certain degree of, but very low, agreement" is what I think should be included in the Discussion part.

p.12, line 203 - Please reconsider the transition "therefore"; I think that the relationship between the preceding and following sentences is not cause and effect.

p.12, line 213 – Among other things, the author defines near misses here; however, it is unclear how this relates to the definition on p.5, line 98. Also, if you are going to provide a definition, please describe it in the Methods section.

Reviewer #2: General, I think the major flaw of your work is methodological issues.

Please provide a structured abstract

Please provide the English translation of Korean version of the questionnaire.

Please provide more information about the validity of Korean version of the questionnaire. Did you calculated judgmental or empirical validity?? The information about validity should be included completely in the manuscript. And if you do not, there is a big flaw in your work.

There was a need to follow the questionnaire validity steps in your work so that we can trust the results of your study.

The questionnaire detail should be completely included in the methods.

I did not find any information about reliability of the questionnaire

Line 98-99: Classification of harm and harm duration

the item A and B is “near miss”?? Is that correct?

You could develop data analysis and perform a regression model between participants characteristics and patient safety event experiences

General Characteristics form should be introduced in the methods.

It is recommended to add the mean and SD for quantitative variables in the General Characteristics of the participants.

Line 130-132: there is need to reworded, It is confusing.

Please summarize key findings of the study In initial part of the discussion.

---

## [Author Response · Author response to Decision Letter 0]

21 Sep 2020

Dear Editor and Reviewer:

We would like to thank the reviewers for their thoughtful review on our manuscript entitled “Differences in the perception of harm assessment among nurses in the patient safety classifica-tion system”. Those comments are all valuable for revising and improving our paper. We agree with their comments and we have revised our manuscript accordingly. 

Please find our responses to the reviewers’ comments below.

Response to comments from editor and reviewers:

Journal Requirements:

1) Please ensure that your manuX-X-SCRIPT meets PLOS ONE's style requirements, including those for file naming.

Response: We have modified our manuscript and file name by referring to the PLOS ONE’s style requirements.

2) In your financial disclosure, please clearly specify whether the funders played any role in the study.

Response: We have specified the role of funders in the financial disclosure.

3) Please provide further details regarding how participants were recruited, including the partic-ipant recruitment date.

Response: We have added how we recruited subjects including the recruitment date. Please refer to ‘Subject of Study’.

4) Please clarify in the methods section where the nurses were recruited from (which hospi-tal/study sites).

Response: We have specifically added from which institution the subjects were recruited. Please refer to “Subject of Study”.

5) Thank you for providing the Korean version of the questionnaire as Supporting Information. Please also include an English copy as Supporting Information.

Response: We would include an English version of the questionnaire.

6) Please amend your current ethics statement to include the full name of the ethics commit-tee/institutional review board(s) that approved your specific study. Once you have amended this/these statement(s) in the Methods section of the manuX-X-SCRIPT, please add the same text to the “Ethics Statement” field of the submission form (via “Edit Submission”).

Response: We added the full name of the ethics committee/institutional review board in the section of “Ethical Considerations” and “Ethics Statement” field of the submission form. 

In the case of Yonsei University Wonju Severance Christian Hospital, one of the institutions that conducted the study, only the approval of the nursing department was required for re-search targeting nurses, so the study was conducted with the approval of the Sangji University IRB, which the principal investigator belongs to. Please refer to “Ethical Considerations”. 

Reviewers' comments:

1) According to the author, this study is based on the fact that accurate assessment of harm by healthcare providers is highly important, and that factors such as lack of adequate training can lead to differences in perceptions of harm. However, there is no citation from the literature on this in the Introduction section.

Response: We have described the differences in perception between system reporters among the factors that have not been reported more accurately by the patient safety incident reporting sys-tem, and added to the Introduction section on the need for education. Please refer to “Introduc-tion”.

2) The relationship between the importance of standardization of classifications and terminolo-gy in incident reporting, which the author mentions in the Introduction and at the beginning of the Discussion, and the differences in nurses' perceptions of events (particularly in terms of im-pact on patients and duration of harm) that this study tried to identify is unclear. I think that this lack of clarity can be attributed to the lack of reference to how differences in nurses' per-ceptions of harm may be detrimental to incident reporting.

Response: We have described in the Discussion section how differences in perception among nurses can be a problem for the patient safety reporting system. Please refer to “Discussion”.

3) This problem also affects the Discussion. The first paragraph of the Discussion is what should be stated in the Introduction, and the second paragraph just redefines what has already been defined in the Methods. In the Discussion, starting from the third paragraph, the results of this study, the following sections have not been adequately discussed: 1) General Characteris-tics of subjects, 2) Subjects' patient safety event experiences, and 3) Consistency Rate among subjects regarding the scenarios. This may be because the purpose of this study is not properly established in the Introduction by citing the literature. Therefore, the authors did not compare the results of this study to the findings of previous studies and show what new findings this study represents in this research area. The paragraph from line 228 suddenly refers to education, but the authors should mention more about the relationship between education and this study in the Introduction.

Response: We have revised the Discussion section as a whole with reference to your valuable comments. Please refer to “Discussion”.

4) In summary, it is necessary to clearly state what this study tries to identify in the field of in-cident reporting and patient safety and to state that this requires a method of determining the rate of agreement between the nurses' perceptions of harm based on the scenarios. The results obtained then need to be discussed in relation to previous studies.

Response: We have clearly described what this study seeks to identify in the areas of patient safety reporting system, and stated that a method for determining the rate of agreement be-tween nurses' perceptions of harm is needed. Please refer to “Discussion”.

Minor comments:

1) p.7, line 139 - I think the author forgot to delete the phrase, "Contents of a patient safety training."

2) p.7, line 141 - Please make sure the phrase "the number of patients." is correct.

3) p.9, line 157 - Please define "patient safety event."

4) p.9, line 163 - It is difficult to understand the phrase, "medication error was the most com-mon answer ".

5) p.10, line 189 - Please use the word for Fleiss' assessment of the Kappa value in the Method section of line 123. "Showing a certain degree of, but very low, agreement" is what I think should be included in the Discussion part.

6) p.12, line 203 - Please reconsider the transition "therefore"; I think that the relationship be-tween the preceding and following sentences is not cause and effect.

7) p.12, line 213 – Among other things, the author defines near misses here; however, it is un-clear how this relates to the definition on p.5, line 98. Also, if you are going to provide a defini-tion, please describe it in the Methods section.

Response: As suggested by the reviewer, we have reviewed carefully the entire manuscript and have modified referring to the minor comments.

Reviewer #2 comments:

1) Please provide a structured abstract.

Response: We have revised the abstract. Please refer to “Abstract”.

2) Please provide the English translation of Korean version of the questionnaire.

Response: We would submit an English version of the questionnaire.

3) Please provide more information about the validity of Korean version of the questionnaire. Did you calculated judgmental or empirical validity?? The information about validity should be included completely in the manuX-X-SCRIPT. And if you do not, there is a big flaw in your work.

4) There was a need to follow the questionnaire validity steps in your work so that we can trust the results of your study.

Response: We developed the Korean version of the questionnaire to evaluate the reliability of harm score assignment entered into the patient safety incidents reporting system by nursing staff. We prepared the questionnaire by referring to the paper below, and the questionnaire was consisted of nine scenarios of patient safety incidents occurring in the clinical field of Korea and required the subjects to evaluate the harm and harm duration of each scenario. Also, after completing the Korean version of the questionnaire, we received reviews and feedback from experts working at clinical and university on whether the scenario of patient safety incidents in the questionnaire reflects the real-world incidents. We have described these details more clearly in the Research Tools, Instrument section.

- Williams T, et al. The reliability of AHRQ Common Format Harm Scales in rating pa-tient safety events. J Patient Saf 2015 Mar;11(1):52-9. 

Please refer to the “English version of the questionnaire” & “Instrument”.

5) The questionnaire detail should be completely included in the methods.

Response: We have revised the details of questionnaire in Instrument section. Please refer to the “Instrument” & “Definition of Harm and Harm Duration of Questionnaire”.

6) I did not find any information about reliability of the questionnaire.

Response: To evaluate the reliability of harm score assignment, we calculated the Fleiss’ kappa which is a statistical measure for assessing agreement among multiple raters assigning categori-cal ratings. Please refer to the “Data Analysis”.

7) Line 98-99: Classification of harm and harm duration.

Response: We described about the classification of harm and harm duration in the Definition of Harm and Harm Duration of Questionnaire section. Please refer to the “Definition of Harm and Harm Duration of Questionnaire”.

8) the item A and B is “near miss”?? Is that correct?

Response: The harm assessment criteria used in our study were developed by NCC MERP, which is currently widely applied in harm assessment in Korean medical institutions. A detailed description of near miss A and B is provided in the Definition of Harm and Harm Duration of Questionnaire section. Please refer to the “Definition of Harm and Harm Duration of Question-naire”

9) You could develop data analysis and perform a regression model between participants char-acteristics and patient safety event experiences.

Response: Thank you for your valuable comment. In this study, we intended to evaluate the consistency of the harm assessment among nurses, and we would like to conduct further re-search to analyze and compare the general characteristics such as patient safety experience and their relevance to the harm assessment.

10) General Characteristics form should be introduced in the methods.

Response: We have added the details of General Characteristics in the Instrument section. Please refer to the “Instrument”.

11) It is recommended to add the mean and SD for quantitative variables in the General Char-acteristics of the participants.

Response: In this study, all of the general characteristics of the subjects were measured as cat-egorical variables, and therefore we calculated the frequency and percent.

12) Line 130-132: there is need to reworded, It is confusing.

Response: We have modified the line 130-132. 

13) Please summarize key findings of the study In initial part of the discussion.

Response: We have modified the Discussion section. Please refer to the “Discussion”.

We sincerely hope the revision will meet with your requirements.

---

## [Decision Letter · Decision Letter 1]

19 Oct 2020

PONE-D-20-19403R1

Differences in the perception of harm assessment among nurses in the patient safety classification system

PLOS ONE

Dear Dr. Shin,

Thank you for submitting your manuscript to PLOS ONE. After careful consideration, we feel that it has merit but does not fully meet PLOS ONE’s publication criteria as it currently stands. Therefore, we invite you to submit a revised version of the manuscript that addresses the points raised during the review process.

We look forward to receiving your revised manuscript.

Kind regards,

Prof, Mojtaba Vaismoradi, PhD, MScN, BScN

Academic Editor

PLOS ONE

Reviewers' comments:

Reviewer #1: Thank you for taking the time to respond to my comments.

Your revision has helped me understand the position of this study in the patient safety field. I think the methods, results, and discussion sections are adequate. However, your conclusions need to be brief and correspond to the introduction section. The conclusions have overlapped slightly with what has been stated in your results and discussion sections.

I would suggest that you instead try to write a straightforward description of the most important findings of this study.

I understand the background and purpose of this study as follows.

　For quality reporting of patient safety event, it is important for the reporter to accurately categorize the degree of patients’ harm.

　In particular, it is important to standardize the assessment of the degree of harm by nurses because they have many opportunities to report it.

　Although some studies have considered the standardization of the assessment of patients’ harm in other countries, the issue remains unclear in Korea.

　Therefore, this study aims to determine whether Korean nurses accurately judge the degree of harm by using a scenario to determine the variability in harm ratings.

　The conclusions for this introduction could be as follows.

　For quality patient safety incident reporting, it is important for healthcare professionals to accurately assess the degree of patients’ harm. This study found that the assessment of the degree of harm by Korean nurses was not standardized. The reason for this variability could be due to the lack of education that takes harm assessment into account. Therefore, training in harm assessment and the development of programs to support this training are both necessary.

　

　If you feel that I have understood exactly what you are trying to say, please refer to my comments.

　If not, please write your own concise conclusions.

Reviewer #2: Dear authors

Thank you for addressing the comment and improvement has been observed. I have two additional short comments.

Line 11: Should be “tool”

In this study for assessing the validity of the developed tool only content validity with 3 experts (without pilot study, face validity, criterion validity) was used. This issue should be mentioned as a main limitation.

---

## [Author Response · Author response to Decision Letter 1]

26 Oct 2020

Dear Editor and Reviewers:

We would like to thank the reviewers for their thoughtful review on our manuscript entitled “Differences in the perception of harm assessment among nurses in the patient safety classifica-tion system”. Those comments are all valuable for revising and improving our paper. We agree with their comments and we have revised our manuscript accordingly. 

Please find our responses to the reviewers’ comments below.

Response to comments from reviewers:

Reviewer #1 comments:

1) Your revision has helped me understand the position of this study in the patient safety field. I think the methods, results, and discussion sections are adequate. However, your conclusions need to be brief and correspond to the introduction section. The conclusions have overlapped slightly with what has been stated in your results and discussion sections. I would suggest that you instead try to write a straightforward description of the most important findings of this study. I understand the background and purpose of this study as follows. For quality reporting of patient safety event, it is important for the reporter to accurately categorize the degree of patients’ harm. In particular, it is important to standardize the assessment of the degree of harm by nurses be-cause they have many opportunities to report it. Although some studies have considered the standardization of the assessment of patients’ harm in other countries, the issue remains unclear in Korea. Therefore, this study aims to determine whether Korean nurses accurately judge the de-gree of harm by using a scenario to determine the variability in harm ratings. The conclusions for this introduction could be as follows. For quality patient safety incident reporting, it is important for healthcare professionals to accurately assess the degree of patients’ harm. This study found that the assessment of the degree of harm by Korean nurses was not standardized. The reason for this variability could be due to the lack of education that takes harm assessment into account. Therefore, training in harm assessment and the development of programs to support this training are both necessary.

Response: Thank you for your valuable and kind comments. We have revised the Conclusion section more clearly with reference to your comments. Please refer to “Conclusion”.

Reviewer #2 comments:

1) Line 11: Should be “tool”. In this study for assessing the validity of the developed tool only content validity with 3 experts (without pilot study, face validity, criterion validity) was used. This issue should be mentioned as a main limitation.

Response: We described about the main limitation on the validity of the questionnaire with reference to your valuable comments in the Discussion section. Please refer to the “Discussion”.

We sincerely hope the revision will meet with your requirements.

---

## [Decision Letter · Decision Letter 2]

19 Nov 2020

PONE-D-20-19403R2

Differences in the perception of harm assessment among nurses in the patient safety classification system

PLOS ONE

Dear Dr. Shin,

Thank you for submitting your manuscript to PLOS ONE. After careful consideration, we feel that it has merit but does not fully meet PLOS ONE’s publication criteria as it currently stands. Therefore, we invite you to submit a revised version of the manuscript that addresses the points raised during the review process.

We look forward to receiving your revised manuscript.

Kind regards,

Prof, Mojtaba Vaismoradi, PhD, MScN, BScN

Academic Editor

PLOS ONE

Reviewers' comments:

Reviewer #1: Thank you for revising your conclusions.

I now have a clearer understanding of the conclusions of your study.

Unfortunately, the conclusions have not been changed in the Abstract, even though the Conclusions have been revised.

Could you please consider whether it is necessary to include the revised conclusion in the Abstract?

I would appreciate it if you could revise it.

Reviewer #2: Dear authors

Thank you for the revised manuscript according comments. In my opinion, no other revisions are needed.

---

## [Author Response · Author response to Decision Letter 2]

20 Nov 2020

Dear Editor and Reviewers:

We would like to thank the reviewers for their thoughtful review on our manuscript entitled “Differences in the perception of harm assessment among nurses in the patient safety classifica-tion system”. Those comments are all valuable for revising and improving our paper. We agree with their comments and we have revised our manuscript accordingly. 

Please find our responses to the reviewers’ comments below.

Response to comments from reviewers:

Reviewer #1 comments:

1) Thank you for revising your conclusions. I now have a clearer understanding of the conclu-sions of your study. Unfortunately, the conclusions have not been changed in the Abstract, even though the Conclusions have been revised. Could you please consider whether it is neces-sary to include the revised conclusion in the Abstract? I would appreciate it if you could revise it.

Response: Thank you for your comments. We should have revised the abstract as well as revised the conclusion. Unfortunately, we missed that point. We have revised the Abstract section with reference to your comments. Please refer to “Abstract”.

We sincerely hope the revision will meet with your requirements.

---

## [Editor Report · Decision Letter 3]

24 Nov 2020

Differences in the perception of harm assessment among nurses in the patient safety classification system

PONE-D-20-19403R3

Dear Dr. Shin,

We’re pleased to inform you that your manuscript has been judged scientifically suitable for publication and will be formally accepted for publication once it meets all outstanding technical requirements.

Kind regards,

Prof, Mojtaba Vaismoradi, PhD, MScN, BScN

Academic Editor

PLOS ONE

---

## [Editor Report · Acceptance letter]

26 Nov 2020

PONE-D-20-19403R3 

Differences in the perception of harm assessment among nurses in the patient safety classification system 

Dear Dr. Shin:

I'm pleased to inform you that your manuscript has been deemed suitable for publication in PLOS ONE. Congratulations! Your manuscript is now with our production department. 

Kind regards, 

on behalf of

Professor Mojtaba Vaismoradi 

Academic Editor

PLOS ONE